# Quantifying the Effect of Grilling and Roasting on the Eating Quality of Lamb Leg Muscles

**DOI:** 10.3390/foods12193609

**Published:** 2023-09-28

**Authors:** Hussein Al-Moadhen, Jarrod C. Lees, Liselotte Pannier, Peter McGilchrist

**Affiliations:** 1School of Environmental and Rural Science, University of New England, Armidale, NSW 2351, Australia; halmoad2@une.edu.au (H.A.-M.);; 2School of Agriculture and Food Sustainability, The University of Queensland, Gatton, QLD 4343, Australia; 3School of Agriculture Sciences, Murdoch University, Murdoch, WA 6150, Australia; l.pannier@murdoch.edu.au

**Keywords:** sheepmeat, shear force, intramuscular fat, connective tissue

## Abstract

Lamb eating quality was measured using untrained consumer sensory panels to determine the difference in intrinsic eating quality scores of grilled and roasted leg cut muscles. The Knuckle, Outside flat, and Topside from both legs of 65 mixed-sex lambs from diverse genetic backgrounds were prepared using alternative grill and roast cook methods. Each sample was eaten by 10 consumers and scored for tenderness, juiciness, flavor, and overall liking. All cuts scored higher (*p* < 0.001) when grilled compared with when roasted for all traits except for Topside tenderness. Grilled Knuckle scored higher than roast Knuckle by 13.6%, 23.9%, 14.4% and 15.8% for tenderness, juiciness, flavor, and overall liking, respectively. The grilled Outside flat scored higher than roast Outside flat by 14.1%, 27.1%, 10.9%, and 14.3% for tenderness, juiciness, flavor, and overall liking, respectively. Finally, grilled Topside scored higher than roast Topside by 21.3%, 7.4%, and 6.6% for juiciness, flavor, and overall liking, respectively. Carcass traits for intramuscular fat and shear force had a significant (*p* < 0.001) effect on all eating quality traits for both grill and roast cuts. Girth rib fat had a significant effect (*p* = 0.01) on tenderness and juiciness (*p* = 0.03) for Outside flat and Topside but had no effect (*p* > 0.05) on Knuckle for both grill and roast. This study identified that specific cooking methods can improve sensory traits for individual cuts and suggests that a cut-by-cook method eating quality model for sheepmeat can therefore increase consumer satisfaction.

## 1. Introduction

Lamb consumption declined in Australia during the twentieth century due to inconsistent products which failed to meet consumer satisfaction requirements [1]. To maintain and strengthen consumer demand, a series of experiments were undertaken to improve sheepmeat quality [2,3,4,5]. In response, the Meat Standards Australia (MSA) system for sheepmeat was created which is a pathways-based total quality management system. This system aimed to deliver guaranteed eating quality to customers using large-scale testing of meat by untrained consumers [6]. Previous work conducted on beef has shown that the cooking method has a profound effect on sensory scores given by consumers [7,8]. Beef research showed that there is a difference in consumer sensory scores (scored out of 100) and grilled scored lower than roasted. For example, grilled *m*. *semimembranosus* (Topside) scored 34.8 and, in contrast, scored 43.4 when roasted [9]. Thus, there is the ability to assign an eating quality score to a group of muscles (primal cut) based on the eating quality prediction of the lowest scoring muscle/portion cut within that primal cut [10]. For example, *m*. *semitendinosus* has a lower eating quality than *m*. *biceps femoris* when grilled [9], and when sold together they make up the Outside flat [11]. The beef MSA prediction model gives the eating quality of the *m*. *semitendinosus* for the Outside flat as it will provide the least favorable eating experience of the two muscles when roasted [10] Lamb eating quality has been measured to develop the cut by cooking method quality prediction system for sheepmeat on a limited range of cuts [5,6,12]. However, the current MSA protocol for roasting and serving whole lamb legs does not take into account the differentiation of individual muscles when roasted. 

The MSA protocol for cooking and serving lamb leg roasts involves cooking and slicing the rolled netted leg roast (Topside, Outside flat, and Knuckle) as a whole. Samples are then portioned from the resulting slice, without regard for a specific muscle or muscle direction. As such, the resulting eating quality scores are a combination of all muscles within the leg. This limits the ability to market individual muscles as roasting cuts. As consumer preferences lead to an increased demand for small, easy-to-cook meals [13], the ability to sell a roast for one or two consumers becomes an attractive opportunity for discerning meat companies. Also, lambs have become bigger over recent decades [14], so portion cut sizes are larger and the cost of whole leg roasts are increasing. This research aims to quantify the effect of the cooking method on the eating quality of three different cuts; *m. semimembranosus*, *adductor femoris* (Topside); *m. biceps femoris* (Outside flat); and *m. rectus femoris*, *vastus lateralis,* and *vastus intermedius* (Knuckle) to further develop the sheepmeat eating quality prediction model. It was hypothesized that the Outside flat, Topside, and Knuckle will have lower consumer eating qualities when cooked using a grill compared to a roast. 

## 2. Material and Methods

### 2.1. Animals and Experiment Design

The experiment was approved by the Animal Ethics Committee of the University of New England (UNE; Authority number: AEC20-001). The research was conducted on lambs from the Meat and Livestock Australia resource flock, which is run at the University of New England Kirby research station (Armidale, NSW, Australia). The lambs were the progeny of industry Terminal, Maternal and Merino sires representing the major production types in the Australian sheep industry mated from Merino ewes. Lambs were born, raised together, grazed on improved pastures, and fed supplementary feeds from weaning when pasture availability was insufficient. Lambs were weighed prior to transport to a commercial abattoir where they were held overnight and slaughtered the following day. Sixty-five (*n* = 65) carcasses were selected for utilization in the current study. Carcasses were subjected to medium-voltage electrical stimulation [15] within 25 min of exsanguination. Carcasses were assigned to a chiller approximately 25 min post-mortem at 4 °C and pH/temperature decline was measured using a WP-80M meter (TPS Pty Ltd., Brendale, Brisbane, QLD, Australia) at the site used by MSA between the 2nd and 5th lumbar vertebrae [16] portion of the left loin (*m. longissimus lumborum LL*) until loin pH < 6.0 or carcasses were <12 °C. The pH meter was calibrated for temperature at 2 °C which aligns with chiller temperature and calibrated for pH before use and every 2 h using pH 4.00 and pH 6.88 buffers at room temperature as per Pearce, Van De Ven, Mudford, Warner, Hocking-Edwards, Jacob, Pethick, and Hopkins [15]. Hot standard carcass weight (HCWT) and girth rib (GR) fat tissue depth (soft tissue depth over the 12th rib, 110 mm from the midline) were also recorded. Carcasses were transported to the University of New England Meat Science laboratory for further measurements and processing 24 h post-mortem. 

Forty-eight hours post-mortem, samples (40 g) for intramuscular fat (IMF%) were collected from the caudal end of the *m. longissimus thoracis* (13th thoracic vertebrae) and the cranial end of the LL (1st lumber vertebrae). The samples were freeze-dried in a Cool Safe Touch 95-15 Pro (Scanvac, laboGene, Lillerød, Denmark), ground, and analyzed using a bench top near-infrared technology using Bruker MAP II Fourier Transform Near-Infrared Spectrometer (Bruker Pty Ltd., Preston, VIC, Australia) calibrated to chloroform Soxhlet fat extraction to estimate chemical fat content as detailed by [17,18,19]. Forty-eight hours post-mortem, an additional 65 g of LL was collected from the 2nd to 4th lumbar vertebrae for shear force at day 5 (SF5) analysis. Samples were vacuum-packed and aged for 5 days post-mortem at 1 °C, and then frozen at −20 °C until subsequent testing [18]. The frozen SF5 samples were cooked in a vacuum-sealed bag in a water bath at 71 °C for 35 min and then cooled in running water for 30 min after cooking [20]. Then, six samples (3–4 cm long, 1 cm^2^ cross-sectional) from each steak were cut with the grain and measured using a Lloyd texture analyzer with a Warner–Bratzler shear blade fitted [21].

### 2.2. Carcass Preparation for Sensory Samples

#### 2.2.1. Sample Preparation

Carcasses were broken down and consumer sensory samples were fabricated for grilling and roasting. Legs were portioned into individual grill cuts from one side, and a Knuckle roast plus a boneless leg roast (Topside and Outside together) from the alternative side. As such, three different cuts; *m. semimembranosus* and *adductor femoris* (Topside-cap off, HAM 5077); *m. biceps femoris* (Outside flat, HAM 5075); and *m. rectus femoris*, *vastus lateralis,* and *vastus intermedius* (Knuckle, HAM 5072) were collected from the left and right legs of each lamb carcass. The sensory cook method was alternated between the left and right leg across all carcasses. Sensory sample fabrication followed the protocol presented by Thompson et al. [5], and this method is currently utilized for all sensory sample fabrication for MSA consumer sensory testing. Briefly, for the grill cook method, cuts were dissected from the carcass, trimmed of subcutaneous and intermuscular fat, and denuded of epimysium. Five steaks were prepared from each cut with dimensions of 15 mm thick × 50 mm long × 50 mm wide across the muscle fibers. The roast cuts were prepared by trimming fat to approximately 4 mm (where fat was present) and boneless leg roasts (containing Topside and Outside attached to each other) were netted using elastic meat netting. All sensory samples were labeled with a unique identifier, vacuum packed, aged for 5 days post-mortem at 1 °C, and frozen at −20 °C until required for untrained consumer sensory testing.

#### 2.2.2. Grill Cooking Method

The grill samples were defrosted in a 2 °C chiller for 24 h prior to grilling. For 30 min prior to grilling, the meat samples were left at room temperature to equilibrate. The samples were grilled using a Silex grill (S-tronic steaker, Silex, Hamburg, Germany), with the top set to a temperature of 180 °C and the bottom plate set to 195 °C. Ten steaks at a time were placed on the grill in a known order to maintain identity. After 2 min and 25 s of cooking with the lid closed, the steaks were removed at an internal temperature ~65 °C (medium doneness) and were then rested for 1.5 min before serving. Each sample was then cut in half and served to ten different consumers who provided sensory scores on each sample. 

#### 2.2.3. Roast Cooking Method

Roast samples were defrosted 24 h before the sensory session at 2 °C and cooked in a Rational electric oven set to a temperature of 160 °C (RATIONAL Australia PTY LTD., Derrimut, VIC, Australia). Roasts were removed when they reached an internal temperature of 65 °C and rested in a thermal insulated box for a minimum of 10 min prior to slicing. Each roast was then sliced into 4 mm thick slices and 10 slice samples were selected from across the whole roast for 10 consumers. All the subcutaneous intermuscular fat and connective tissue were removed and slices were trimmed to approximately 5 cm wide by 5 cm long. Then, samples were placed in stainless steel pans (Bain-marie) with a temperature of 55 °C maintained until serving. 

#### 2.2.4. Consumer Survey

All sensory data presented here were collected with the approval of the University of New England Human Ethics committee (HE17-253). Briefly, ten untrained consumers assessed each grill and roast cut for tenderness, juiciness, flavor, and overall liking using a visual analog scale from 0 to 100. The analogue scale is anchored by dislike extremely at 0 and like extremely at 100 for flavor and overall liking, while tenderness is anchored by not tender at 0 and very tender at 100 and juiciness is anchored by not juicy at 0 and very juicy at 100 [1]. Each member of the test panel consumed a common starter sample of average quality to set their palette, followed by six experimental samples.

The experimental samples were served and allocated to untrained consumers using a 6 × 6 Latin square design to avoid any sensory sample order bias between consumers [10] and ensure that each sample in the Latin square was eaten before and after every other sample, which consisted of 60 unique consumers testing 36 cuts, finally obtaining 10 consumer responses per cut. The 10 consumer responses were averaged to provide a mean consumer sensory score for each cut by the cook for each animal for tenderness, juiciness, flavor, and overall liking to take into account consumer variability. For grill samples, 324 untrained consumers were utilized and 325 different untrained consumers were utilized to test the roast samples. The number of samples and the mean raw sensory scores are presented in Table 1.

### 2.3. Statistical Analysis

The consumer meat eating quality score (MQ4) was calculated based on the combined mean consumer score of tenderness, juiciness, flavor, and overall liking for each cut. This study weighted the sensory scores 0.3 × tenderness, 0.1 × juiciness, 0.3 × flavor, and 0.3 × overall liking to estimate the MQ4 score for each cut [6,10].

The effect of the cut and cook on meat sensory scores (tenderness, juiciness, flavor, overall liking, and MQ4) was analyzed using the lmer package for linear mixed effects models in R [22]. Statistical significance terms were identified as *p* value equal or less than 0.05. The base model for each sensory trait included cut (Knuckle, Outside flat, and Topside) and cooking method (roast and grill) as fixed effects, and the associated interaction. Animal identification was used as a random term. Consumer session was included as a random term; however, it did not have a significant impact on the least square means so was subsequently removed from the models along with all non-significant (*p* > 0.05) terms in a stepwise manner. 

The effect of carcass traits (Table 2), including HCWT, IMF%, SF5, and GR fat, on meat sensory scores were analyzed individually and in combination using linear effect models in R [23], including their interaction with cooking method and muscle type. These models included all relevant first-order interactions between fixed effects and covariates, with their linear and quadratic effects for each covariate and the non-significant (*p* > 0.05) terms were removed in a stepwise manner. Animal identification was not used as a random term in the analysis when carcass traits were added because there is a strong positive correlation between C-site fat and GR fat. Both traits were initially added in the models, which showed that GR fat had the larger F value and explained more variation in eating quality traits so it was used as the fat descriptor of the carcasses in the analysis.

## 3. Results

### 3.1. The Effect of Cooking Method and Lamb Leg Cuts on Eating Quality

Consumer sensory scores for tenderness, juiciness, flavor, and overall liking varied between grill and roast for the three cuts tested in this study (Table 1). The base models described 30%, 26%, 31%, and 30% of the total variance in tenderness, juiciness, flavor, and overall liking sensory scores, respectively. There was a significant effect of cooking method on sensory scores which varied by cut on sensory scores (*p* < 0.001, Figure 1, Table 1).

The grilled Knuckle scored higher than the roast Knuckle by 13.6%, 23.9%, 14.4%, 15.8%, and 15.6% for tenderness, juiciness, flavor, overall liking, and MQ4, respectively. The grilled Outside flat scored higher than roast Outside flat by 14.1%, 27.1%, 10.9%, 14.3%, and 14.5% for tenderness, juiciness, flavor, overall liking, and MQ4, respectively. The grilled scored higher by 21.3%, 7.4%, 6.6% and 6.9% for juiciness, flavor, overall liking, and MQ4, respectively, when grilled compared to roasted. However, tenderness was not different between Topside grill and roast (*p* > 0.05).

Within a cooking method, variation between cuts existed. Tenderness of the Topside grill was the lowest at 49.3 ± 1.36, which was 11.2 points and 18.2 points lower than the tenderness score of the grilled Outside flat and Knuckle, respectively (*p* < 0.001). The tenderness of the grilled Outside flat was 7 points lower than the Knuckle (*p* < 0.001). The Topside also scored lowest for tenderness when roasted, scoring 48.2 ± 1.36 points, which was significantly lower than the tenderness of the roasted Knuckle by 10.1 points (*p* = 0.001). On the other hand, the tenderness of the roasted Topside did not differ from the Outside flat; however, the tenderness of the roasted Outside flat was 6.3 points lower than the roasted Knuckle’s tenderness (*p* = 0.001) (Figure 1).

Juiciness of the Topside grill was the lowest at 54.1 ± 1.27, which was 8.2 points and 13.8 points lower than the juiciness score of the grilled Outside flat and Knuckle, respectively (*p* < 0.001). The juiciness of the grilled Outside flat was 5.5 points lower than the Knuckle (*p* = 0.004). The Topside also scored the lowest for juiciness when roasted, scoring 42.6 ± 1.27 points, which was significantly lower than the tenderness of the roasted Knuckle by 9.1 points (*p* = 0.001). The juiciness of the roasted Topside did not differ from the Outside flat; however, the juiciness of the roasted Outside flat was 6.2 points lower than Knuckle juiciness (*p* = 0.001) (Figure 1).

Flavor of the Topside grill was the lowest at 55.8 ± 1.13, which was 5.8 points and 9.5 points lower than the flavor score of the grilled Outside flat and Knuckle, respectively (*p* < 0.001). The flavor of the grilled Outside flat was 3.7 points lower than the Knuckle (65.2 ± 1.14, *p* = 0.032). The Topside also scored the lowest for flavor when roasted scoring 51.7 ± 1.13 points, which was significantly lower than the flavor of the roasted Knuckle, by 4.13 points (*p* = 0.013). The flavor of the roasted Topside did not differ from the Outside flat and the Outside flat did not differ in flavor from the Knuckle (Figure 1). 

Overall liking of the Topside grill was the lowest at 53.3 ± 1.22, which was 8.4 points and 13 points lower than the overall liking score of the grilled Outside flat and Knuckle, respectively (*p* < 0.001). The overall liking of the grilled Outside flat was 4.8 points lower than the Knuckle (*p* = 0.007). The Topside also scored the lowest for overall liking when roasted, scoring 49.8 ± 1.22 points, which was significantly lower than the overall liking of the roasted Knuckle by 6.2 points (*p* = 0.001). The overall liking of the roasted Topside did not differ from the outside flat and the Outside flat did not differ from the Knuckle (Figure 1). 

The MQ4 score of the Topside grill was lowest at 52.9 ± 1.17, which was 8.4 points and 13.7 points lower than the MQ4 score of the grilled Outside flat and Knuckle, respectively (*p* < 0.001). The MQ4 of the grilled Outside flat was 5.3 points lower than the Knuckle (*p* = 0.002). The Topside also scored the lowest for MQ4 when roasted, scoring 49.2 ± 1.17 points, which was significantly lower than the MQ4 of the roasted Knuckle by 7 points (*p* = 0.001). The MQ4 of the roasted Topside did not differ from the Outside flat (*p* = 0.07) and the Outside flat was significantly (*p* = 0.03) different in MQ4 to the Knuckle (*p* = 0.04) by 3.75 points (Figure 1).

### 3.2. Association between Carcass Traits and Eating Quality

The unadjusted means, standard deviations, minimum and maximum values of all carcass traits measured on the 65 lambs and utilized in the statistical models are shown in Table 2. They were heavy export lambs with an average HCWT of 27.5 kg and a range from 22 to 32.8 kg. The lambs were also well finished with an average of 20.5 mm GR depth and had a high average IMF of 6.41% and a range from 3.8 to 10.8%. The average SF5 value of 24.3 N with a range from 12.9 to 43.6 reflected the high IMF percentages in these lambs.

There was a significant effect of carcass traits for IMF%, GR fat, and SF5 on the sensory scores, which varied for the different cuts by cooking methods (*p* < 0.001, Figure 2). However, HCWT had no impact on tenderness, juiciness, flavor, and overall liking or MQ4 for any of the cuts (Knuckle, Outside flat, and Topside) or cooking method (grill and roast) tested. The interaction between any of the carcass traits and cut or cook method or the second order interaction was also not significant. There was a significant (*p* < 0.001) positive effect of IMF% on Knuckle and Outside flat for grill and roast, and grilled Topside (Table 3). As IMF% increased from 4% to 10%, the grilled Knuckle’s tenderness, juiciness, flavor, overall liking, and MQ4 increased by 16.7%, 13.9%, 16.5%, 19.6%, and 6.6%, and roasted Knuckle increased by 16.3%, 15.2%, 15.2%, 14.7%, and 7.9% for tenderness, juiciness, flavor, overall liking, and MQ4, respectively. Grilled Outside flat increased by 7.8%, 16.4%, 13.9%, 13.5%, and 7.2% for tenderness, juiciness, flavor, overall liking, and MQ4, respectively, and roasted Outside flat increased by 5.7%, 18.9%, 11.5%, 14.7%, and 8.5% for tenderness, juiciness, flavor, overall liking, and MQ4 when IMF% increased from 4% to 10%. The grilled Topside tenderness, juiciness, flavor, overall liking, and MQ4 increased by 9.7%, 8.4%, 4.8%, 9.5%, and 8.4%, respectively, across the 6% IMF range, whereas roasted Topside increased by 5.9%, 6.7%, and 8.8% for tenderness, juiciness, and MQ4, respectively. In addition, there was no effect (*p* > 0.05) of IMF% on flavor and overall liking of the roast Topside.

There was a significant (*p* < 0.001) negative effect of GR fat on the tenderness and juiciness of the grilled and roasted Outside flat and Topside. As GR fat increased from 10 mm to 30 mm, tenderness decreased by 15.3% and 19.2% in the grilled Outside flat and Topside and by 10.5% and 12.4% for roasted Outside flat and Topside. Juiciness decreased by 12% and 12.9% for grilled Outside flat and Topside and 15.4% and 15.5% for roasted Outside flat and Topside as GR fat increased from 10 mm to 30 mm. However, GR fat had no significant effect on flavor, overall liking, or MQ4 for any cut (Figure 2, *p* > 0.05). In addition, there was no GR fat effect on tenderness or juiciness of the Knuckle when grilled or roasted (*p* > 0.001).

There was a significant (*p* < 0.001) negative effect of loin SF5 on tenderness, juiciness, flavor, overall liking, and MQ4 score for both grilled and roasted Knuckle, Outside flat, and Topside (Figure 2). The largest impact of SF5 was seen in the Topside for all sensory traits for both cooking methods. Tenderness, juiciness, flavor, overall liking, and MQ4 of grilled Topside decreased by 34.6%, 23.7%, 23.1%, 29.4%, and 17.9%, respectively, as SF5 increased from 10 N to 45 N (*p* < 0.001). Across this SF5 range from 10 N to 45 N, roasted Topside decreased by 35.4% for tenderness, 35.1% for juiciness, 20.5% for flavor, 29.8% for overall liking, and 19.1% for MQ4 (*p* < 0.001). Additionally, as SF5 increased from 10 N to 45 N, the tenderness, juiciness, flavor, overall liking, and MQ4 of grilled Knuckle decreased by 13%, 6.6%, 9.6%, 13.7%, and 14.4%, respectively (*p* < 0.001). Roasted Knuckle decreased by 15.1% for tenderness, 14.4% for juiciness, 6.6% for flavor, 14.8% for overall liking, and 16.9% for MQ4 as SF5 increased from 10 N to 45 N (*p* < 0.001). The tenderness, juiciness, and MQ4 of grilled Outside flat decreased by 23.2%, 11.5%, and 15.6% while both flavor and overall liking decreased by 17.6%, respectively, as SF5 increased from 10 N to 45 N (*p* < 0.001). Roasted Outside flat decreased by 26.5% for tenderness, 21.5% for juiciness, 15.7% for flavor, 8.8% for overall liking, and 18% for MQ4 as SF5 increased from 10 N to 45 N (*p* < 0.001).

## 4. Discussion

There was a consistent effect of the cooking method on consumer scores across muscle type cuts tested, with grilled muscle cuts yielding higher sensory scores than roasted muscle cuts, which contradicts the initial hypothesis. Watson et al. [9] found that individual beef muscles cooked using different methods had varying sensory scores, which is supported by this study for sheepmeat cuts. However, in beef, there were greater variations within grilled cuts than roasted cuts. Furthermore, the MSA beef eating quality model predicts that roasted Knuckle, Outside flat, and Topside have a higher tenderness, juiciness, flavor, and overall liking score than when grilled [6], which is in contrast with this experiment. This outcome corroborates the work of Ngapo, Gariepy, [24] who noted that the eating quality of the two species was distinctly different. These differences between cooking methods for lamb and beef might be due to differences in age at slaughter, with beef typically slaughtered at an older age than lamb. Lamb age in this study ranged from 345 to 358 days, in contrast with beef, ranging from 356 to 548 days in Australian practice [3,25]. Slaughter age or weight had no impact on the grill and roast results within the lamb experiment because paired samples came from the left and right legs of each lamb. Hence, the differences between the grill and roast samples from the lambs cannot be explained by these factors. In addition, the size of the animals and muscles [26,27] and/or nutrition, which causes diverse differences in beef, in contrast with lamb [28], may be a possible reason. The other cause may be because grilled lamb samples are 15 mm thick, while beef steaks are 25 mm thick [9]. This reduced thickness in the lamb samples and their younger age at slaughter might make for an improved consumer experience. The underlying mechanisms causing the opposite effects in beef and lamb are not clear but could be a target for future research.

The MSA beef and sheepmeat eating quality prediction models are designed to determine the eating quality of a cut for relevant cooking methods using a number of animal and processing inputs. The reason for this approach is due to the fact that the amount of connective tissue can impact the sensory score [29] and, therefore, consumer preferences, which are directly linked to repurchasing behavior and willingness to pay [30]. Previous research has shown that grilling muscles with low amounts of connective tissue resulted in higher sensory scores [31,32], and the difference between scores for the same muscle cooked in different ways is minimal. However, roasting muscles with high levels of connective tissue, like the *m. gluteus medius*, can produce higher sensory scores when compared with grilling [6] because cooking for a long time can increase the disruption and breakdown of connective tissue [33]. In contrast with beef eating quality, this experiment showed that the eating quality scores for grilled lamb cuts scored higher for all three muscles than roasted. Again, the 15 mm slice thickness for the lamb grills may have reduced the chewing resistance for consumers, providing a difference in relationship between the cooking methods when compared to what is experienced in beef.

Tenderness is one of the most important sensory properties of bovine [33] and ovine meat [3] and it is the result of three components; proteolysis post-mortem, sarcomere length, and the amount of collagen [34]. Tenderness corresponds to the ease of mastication during meat consumption [35]. All of the lamb carcasses in this experiment were Achilles hung, which impacts the sarcomere lengths of different muscles due to the muscles’ capacity for shortening to occur prior to rigor mortis. The Topside and Outside have been shown to shorten to a greater extent than the Knuckle [36,37]. This might be further exacerbated by the chilling rate. Bouton et al. [38] showed that the sarcomere length of the Knuckle, Outside, and Topside were significantly affected by chiller temperature and had shorter sarcomere lengths when chilled quickly, leading to tougher meat. The Topside is located at the top of the hind quarter and could be closer to the chiller cooling airflow, increasing the chill rate [39] which increases the risk of cold shortening [37]. In contrast, the Knuckle is located under the Outside flat and deeper down the leg, with a reduced airflow which might reduce its risk of cold shortening. Chill rate and sarcomere length are proposed as possible mechanisms for the Knuckle having a higher tenderness. 

There is a known negative relationship between collagen concentration and sensory score [33]. Muscles that are proportionally higher in collagen content tend to be tougher [40], as evidenced by Hastie et al. [41], who found that the increase in collagen content from loin to Topside reduced all sensory score traits (tenderness, juiciness, flavor, and overall liking). In the current study, the Knuckle scored higher for tenderness compared with the Outside flat and Topside. The differences between muscles might be caused by connective tissue content and collagen solubility [42]. Tschirhart-Hoelscher et al. [43] showed that the total collagen between the *m. rectus femoris* and the *m. semimembranosis* was not different; however, it is well established that muscle collagen solubility can also be impacted by cooking methods [44]. The quantity of soluble collagen may differ between the Knuckle and Topside, which has the capacity to change sensory scores [45,46]. In this experiment, lamb Knuckle might have higher tenderness scores across both grill and roast due to having less collagen at the time of consumption. 

The differences between cooking methods might be related to water-holding capacity [47] and lipids discharged during mastication as both enhance meat juiciness [35]. Juiciness is related to the dryness characteristic of meat during chewing, which can be impacted by the amount of water released during the first bites that is induced by the rapid release of fluid from the meat [48] and the influence of lipids on the secretion of saliva [49]. However, juiciness is not only dependent on the characteristics of meat but also on physiological factors tightly linked to the consumer. It is well known that lipids can affect meat flavor during cooking, and it is linked to cooking temperature [44]. The more intense heating in grilling and roasting leads to the generation of unstable compounds caused by lipid oxidation [50], resulting in a reduction in consumers’ preferences [51,52]. This aligns well with this study, which has shown how cooking processes including temperature and time had a critical influence on the final edible product. The current MSA cooking protocol for grill cooking is to serve as medium doneness and be served in a timely manner. In comparison, a roast needs to reach an internal temperature of 65 °C before removal from the oven and the meat needs to be rested for a minimum of 10 min. Once sliced, the roast meat is retained in bain-maries in a commercial water bath at 55 °C until service (up to 1:5 h). This allows for lipid and moisture loss during cooking and prior to consumption of roast samples, which would impact the sensory outcomes. 

The fat in meat plays an important role in overall meat palatability, and IMF% positively influences eating quality traits such as tenderness, flavor, juiciness, and overall liking [53]. The general relationship between IMF% and eating quality is positive. Generally, as the content of IMF% increases, the eating quality increases. Hopkins et al. [45] reported that increasing IMF% was shown to increase the overall liking score. In addition, Pannier et al. [12] presented that IMF% is the strongest positive driver for eating quality scores and has a direct effect on tenderness, juiciness, flavor, and overall liking of lamb. The muscle structure changes when IMF% is at higher levels which in turn impacts on the tenderness and subsequent sensory effects a [54]. When IMF% levels are <8%, which is generally the case for lamb, its impact is possibly driven through enhanced tenderness, juiciness, and flavor.

The study by Pannier et al. [12] showed that an increase in lamb loin IMF% level of 4.5% increased the sensory scores of grilled Topside by 6.7%, 6.6%, and 5.4% for juiciness, overall liking, and flavor, respectively. In comparison, the current study predicted that an increase from 4% to 10% IMF% was associated with increased eating quality levels of 9.7%, 8.4%, 4.8%, and 9.5% of grilled Topside for tenderness, juiciness, flavor, and overall liking, respectively. This experiment targeted three hindquarter cuts across two different cooking methods, while the study of Pannier et al. [12] utilized only grilled loin and topside. In this study, the most significant effect of IMF% is associated with tenderness and, then, overall liking. This is different from the findings by Pannier et al. [12], which illustrate that the strongest association with IMF% was observed for juiciness and flavor in the grilled topside. In addition, the strongest effect of IMF% for grilled Knuckle was associated with overall liking and tenderness, and was 19.6% and 16.7%, while juiciness and flavor scored 13.9% and 16.5%, respectively, but had the largest effect for Outside flat juiciness, at 16.4% and tenderness at 7.8%. Unfortunately, this experiment did not measure IMF% for each cut due to the muscle size and samples needed but it would be suitable for further research. Using loin IMF% as a predictor of carcass IMF% and predicting its effect on the sensory score of the leg muscles has some limitations because the ratio is not always close to one [55]. 

Warner–Bratzler shear force of LL measured 5 days post mortem had a significant negative relationship with tenderness for all cuts, whereas there were minimal negative effects on other sensory traits for the Knuckle, Outside flat, and Topside. This finding aligns well with other studies that show lower sensory scores when SF5 increases [45,56]. The effect of SF5 is associated with the changing level of IMF% and lean meat yield, which is a reflection of HCWT; it is known that animals with high meat yield had high SF5 [57] and lower sensory scores [12]. In this experiment, even when SF5 and IMF% were included in the model together, both still had a significant effect on the sensory outcomes, indicating that their effect is different, yet complimentary, to each other.

## 5. Conclusions

These findings demonstrate the huge impact that cooking methods and different cuts have on the tenderness, juiciness, and flavor of lamb leg muscles. Grilled cuts scored significantly higher than roasted cuts for all three muscles across all sensory traits. The results of this study provide the opportunity to develop better eating quality prediction models for specific cuts with changing cooking methods. The MSA sheepmeat eating quality model will be enhanced by these results, allowing greater extraction of value for cuts for specific cooking methods. The results also showed that all sensory scores for all cuts by different cooking methods were positively impacted by higher IMF% and negatively impacted by higher shear force. GR fat had a positive but smaller impact on the tenderness and juiciness of the grilled and roasted Outside flat and Topside only. These results support the immense importance of selecting animals with higher IMF% and lower shear force to generate a greater eating quality experience for consumers, which ultimately leads to greater extraction of value for the supply chain. The mechanisms causing the differences between cuts are unclear, but might be caused by the collagen content, IMF%, or other intrinsic factors such as sarcomere lengths. These factors can impact the cooking process under different cooking methods and, therefore, impact sensory scores. Further research is required to determine the influence of other cooking methods on a wider variety of cuts to understand the sensory attributes which are impacted by cooking method. In conclusion, hind leg cuts that have been traditionally roasted from lambs have an improved eating quality when grilled.

## Figures and Tables

**Figure 1 foods-12-03609-f001:**
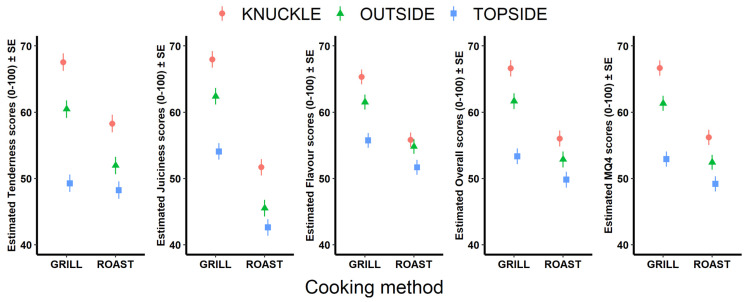
Estimated marginal means (±S.E.) for tenderness (0–100), juiciness (0–100), flavor (0–100), overall liking (0–100), and MQ4 score (0–100) from the base models for consumer sensory scores of the Knuckle, Outside flat, and Topside cooked using the grill and roast methods.

**Figure 2 foods-12-03609-f002:**
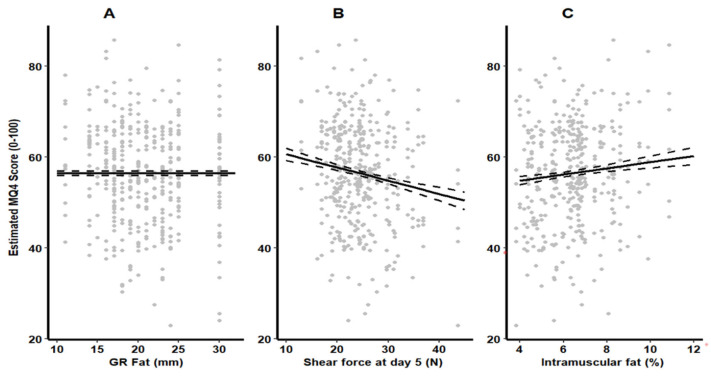
Estimated marginal means from the base model for the effect of (**A**) GR fat (mm), (**B**) shear force at day 5 (N) and (**C**) Intramuscular fat (%) for both grill and roast methods on the meat quality score (MQ4) calculated using the following weightings (30% tenderness, 10% juiciness, 30% flavor, and 30% overall liking).

**Table 1 foods-12-03609-t001:** The number of samples, along with averages, standard deviations, minimum and maximum scores for tenderness, juiciness, flavor, overall liking, and MQ4 scores for the Knuckle, Outside flat, and Topside cooked using grill and roast method.

	Knuckle		Outside Flat		Topside	
	GRILL	ROAST	GRILL	ROAST	GRILL	ROAST
	(*n* = 64)	(*n* = 65)	(*n* = 65)	(*n* = 65)	(*n* = 65)	(*n* = 65)
Tenderness						
Average ± SD	67.6 ± 10.8	58.3 ± 9.98	60.5 ± 9.59	52.0 ± 11.3	49.3 ± 10.8	48.2 ± 12.8
Minimum	45.3	35.5	38.5	29.9	23.5	19.3
Maximum	90.6	84.9	78.8	80.5	69.8	79.3
Juiciness						
Average ± SD	68.0 ± 7.60	51.7 ± 10.8	62.4 ± 8.09	45.5 ± 11.3	54.1 ± 8.87	42.6 ± 13.4
Minimum	53.5	29.9	42.3	25.4	29.2	14.6
Maximum	86.3	75.3	83.6	70.7	73.7	72.1
Flavor						
Average ± SD	65.3 ± 8.48	55.8 ± 9.35	61.5 ± 7.77	54.8 ± 9.52	55.8 ± 7.89	51.7 ± 11.3
Minimum	42.8	30.7	44.2	33.7	39.2	23.6
Maximum	82.9	78.4	75.1	78.2	75.2	75.5
Overall liking						
Average ± SD	66.6 ± 9.14	56.0 ± 9.40	61.7 ± 8.36	52.9 ± 10.1	53.3 ± 9.55	49.8 ± 12.2
Minimum	42.8	34.3	40.6	33.1	35.3	22.1
Maximum	86.1	77.4	79.1	76.0	69.5	77.7
MQ4 *						
Average ± SD	66.7 ± 8.65	56.2 ± 9.03	61.3 ± 7.84	52.4 ± 9.82	52.9 ± 8.88	49.2 ± 11.8
Minimum	45.8	34.1	44.6	31.7	33.6	22.9
Maximum	85.7	79.3	75.4	75.8	70.6	77.0

* The MQ4 was calculated using the weight for each sensory score as listed (30% tenderness, 10% juiciness, 30% flavor, and 30% overall liking).

**Table 2 foods-12-03609-t002:** Descriptive statistics of the carcass traits for shear force (SF5), intramuscular fat percentage (IMF%), girth rib tissue depth (GR fat), and hot standard carcass weight (HCWT). Data are represented for 65 lambs.

Traits	SF5 (N)	IMF (%)	GR Fat (mm)	HCWT (kg)
Average ± SD	24.3 ± 5.48	6.41 ± 1.49	20.5 ± 4.71	27.5 ± 2.89
Minimum	12.9	3.80	11.0	22.0
Maximum	43.6	10.8	30.0	32.8

**Table 3 foods-12-03609-t003:** Descriptive statistics for the response of sensory traits base mix models score from two cooking methods (grill and roast) and predictors broken down by three muscles (Knuckle, Outside flat, and Topside).

	Tenderness		Juiciness		Flavor		Overall		MQ4	
Predictors	Estimates	*p*	Estimates	*p*	Estimates	*p*	Estimates	*p*	Estimates	*p*
(Intercept)	67.48	<0.001	67.9	<0.001	65.22	<0.001	66.53	<0.001	66.56	<0.001
Knuckle	Reference		Reference		Reference		Reference		Reference	
Outside flat	−7.01	<0.001	−5.5	0.001	−3.7	0.012	−4.87	0.002	−5.22	0.001
Topside	−18.22	<0.001	−13.83	<0.001	−9.47	<0.001	−13.2	<0.001	−13.64	<0.001
Grill	Reference		Reference		Reference		Reference		Reference	
Outside: Roast	0.7	0.777	−0.67	0.78	2.71	0.189	1.72	0.442	1.47	0.49
Roast	−9.22	<0.001	−16.22	<0.001	−9.42	<0.001	−10.51	<0.001	−10.36	<0.001
Topside: Roast	8.19	0.001	4.75	0.048	5.34	0.01	6.99	0.002	6.63	0.002
Random Effects										
σ^2^	99.65		92.93		68.79		80.98		73.27	
τ00	19.92		11.81		14.88		16.43		15.52	
ICC	0.17		0.11		0.18		0.17		0.17	
*n*	65		65		65		65		65	
Observations	389		389		389		389		389	
Marginal R^2^/Conditional R^2^	0.280/0.400		0.428/0.493		0.196/0.339		0.250/0.376		0.280/0.406	

σ^2^ = represents the variance of that random variable and 2 is superscript, τ = t-statistic, *n* = number of animals, R^2^ = Coefficient of determination

## Data Availability

Data is contained within the article.

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
