# Peer review of "Quantifying the Effect of Grilling and Roasting on the Eating Quality of Lamb Leg Muscles"

_foods, 2023, doi:10.3390/foods12193609_

Round 1

Reviewer 1 Report

Overall, the quality of this manuscript is ordinary, with no innovations. The text is not adequately presented and the sensory evaluations are lacking, as well as some of the experiments and data to support them.

1.        Line 99. Sensory evaluation includes color, flavor, taste, and texture. This manuscript is not adequately prepared and lacks a complete sensory evaluation process and visual images. It does not allow the reader to feel the importance of the effects of different cooking methods and different cuts on lamb shanks. Other indicators are also important factors in measuring the quality of a product and need to be added.

2.        Line 144. Sensory evaluation of the crowd is very important and is not presented separately here. Sensory evaluations are categorized as expert, consumer, inexperienced, experienced, and trained. Is the range of people from whom data was collected for this study targeted at the inexperienced? This is a group of sensory evaluators who only evaluate the level of enjoyment and acceptance of the product, but this group of people is not representative of the type of consumer and can generally only perform the sensory evaluation in a small laboratory setting. The number of sensory evaluators is small and they are not representative.

3.        Line 221. The lack of part of the experimental data, should be GC-MS to determine the volatile aroma to support the evidence of taste evaluation, can not only rely on subjective feeling evaluation of taste good or bad, as which type of meat aroma substance content is higher (alcohol, aldehyde, acid, and so on), need to add the electronic nose, electronic tongue and other equipment to determine the taste data, can not rely on the sensory evaluation score of 10 persons alone to distinguish between which type of meat is good or bad.

4.        Line 189-290. Combined with the analysis results of 3.1-3.2, tenderness can correspond to shear force and intramuscular fat content. I think juiciness can be added as part of the cooking loss test.

5.        Line 189. What does muscle type mean here? Muscle types are generally referred to as Type I muscle fibers, Type IIA muscle fibers, and Type IIB muscle fibers. It is not clear which fiber type predominates in the three parts of the material.

6.        Line 407. Speculation about principles by summarizing the literature may be added to the discussion, but the language of the conclusion should be rigorous, concise, accurate, and logical. Words such as "probably," "maybe," "perhaps," and "could be" should not be used. The use of these words will make the reader doubt the authenticity and scientific nature of the findings. The conclusions may be arranged in order of importance of the results.

7.        The abbreviation that first appears in the abstract is not shown by its full name

8.        There is no mention of animal ethics-related standards in the article, and whether there is documentation to support them

9.        Line 59-60: There doesn't seem to be much difference between meat cooked on a grill and a roast

10.     Line 68: What is the relationship between lamb supplementation and this trial?

11.     Results: Benchtop NIR technique for analysing chemical fat content was not present in the results

需要对英语进行少量编辑

Author Response

Reviewer #1

S.N

Comment/s

Response to comments and changes made

1.

Line 99. Sensory evaluation includes color, flavor, taste, and texture. This manuscript is not adequately prepared and lacks a complete sensory evaluation process and visual images. It does not allow the reader to feel the importance of the effects of different cooking methods and different cuts on lamb shanks. Other indicators are also important factors in measuring the quality of a product and need to be added.

The authors agree that meat colour, fat colour and meat texture are important however this experiment focused on consumer appreciation of tenderness, flavour (taste), juiciness and overall liking of cooked lamb. Hence the importance of meat and fat colour is negated because of the cooking process prior to consumption. Furthermore, visual images are never part of the Meat Standards Australia (MSA) protocol.

Lamb shanks are an important cut on the lamb leg however are better suited to the stew or slow cook method, hence were not tested using the grill and roast cook methods because a comparison could not be made.

2

Line 144. Sensory evaluation of the crowd is very important and is not presented separately here. Sensory evaluations are categorized as expert, consumer, inexperienced, experienced, and trained. Is the range of people from whom data was collected for this study targeted at the inexperienced? This is a group of sensory evaluators who only evaluate the level of enjoyment and acceptance of the product, but this group of people is not representative of the type of consumer and can generally only perform the sensory evaluation in a small laboratory setting. The number of sensory evaluators is small and they are not representative.

Numbers of consumers utilised in the study has been added to the methods section. This research utilized 324 untrained consumers for the grill cook method and 325 different untrained consumers were utilized to test the roast samples. The consumer testing was conducted as per Watson 2008 which is the basis to the Meat Standards Australia protocol. Consumers are screened to represent the population of consumers that purchase and eat lamb in Australia. Trained consumer panels were not used. Consumers ranged in age, profession, income, frequency of red meat consumption

3

Line 221. The lack of part of the experimental data, should be GC-MS to determine the volatile aroma to support the evidence of taste evaluation, can not only rely on subjective feeling evaluation of taste good or bad, as which type of meat aroma substance content is higher (alcohol, aldehyde, acid, and so on), need to add the electronic nose, electronic tongue and other equipment to determine the taste data, can not rely on the sensory evaluation score of 10 persons alone to distinguish between which type of meat is good or bad.

Thank you for the suggestion however this experiment or manuscript didn’t include volatile aroma, as the focus was on the differences between cooking method and not the mechanisms causing possible effects. Please note that due to the size of lamb muscles, each muscle were eaten by 10 consumers. Our work was part of a larger longitudinal study looking at the impact of different traits on the consumer sensory experience. It is commonly held belief that the human’s ability to determine what they like and don’t like is better than that of objective measurement technologies, because it is the consumers that pay for and appreciate product.

4

Line 189-290. Combined with the analysis results of 3.1-3.2, tenderness can correspond to shear force and intramuscular fat content. I think juiciness can be added as part of the cooking loss test.

Yes tenderness is indeed affected by shear force (SF5) and IMF%, however because left and right legs of the same animal were evaluated as part of this study, including animal ID as a random term in the analysis accounted for differences in SF5 and IMF%. However the carcass traits were also included in the analysis individually as outlined in the 2nd paragraph of the statistical analysis section. Cooking loss was not captured for each sample because there was 5 separate steaks for each grilled muscle and roast samples were grilled in combination with each other so cook loss per cut is confounded in this method.  

5

Line 189. What does muscle type mean here? Muscle types are generally referred to as Type I muscle fibers, Type IIA muscle fibers, and Type IIB muscle fibers. It is not clear which fiber type predominates in the three parts of the material.

Accepted; changed to lamb leg cuts

6

Line 407. Speculation about principles by summarizing the literature may be added to the discussion, but the language of the conclusion should be rigorous, concise, accurate, and logical. Words such as "probably," "maybe," "perhaps," and "could be" should not be used. The use of these words will make the reader doubt the authenticity and scientific nature of the findings. The conclusions may be arranged in order of importance of the results.

Accepted. Conclusion changed significantly.

7

The abbreviation that first appears in the abstract is not shown by its full name

Accepted. The abbreviation is changed with the full name (line 19)

8

There is no mention of animal ethics-related standards in the article, and whether there is documentation to support them

Ethics approval was provided and this information was added at the start of the methods along with the human ethics approval number for the experiment.

9

Line 59-60: There doesn't seem to be much difference between meat cooked on a grill and a roast

This was hypothesis of the experiment, not an actual finding of the study. No changes made.

10

Line 68: What is the relationship between lamb supplementation and this trial?

There is no relation. This information was included to describe how animals were fed during a time of pasture shortage. 

All lambs came from 1 cohort that was born, raised and slaughtered together so had the same nutritional treatment throughout life.

11

Results: Benchtop NIR technique for analysing chemical fat content was not present in the results

The authors are unsure exactly what this comment relates to. The method for the NIR determination of IMF% is outlined extensively in the methods with references to other published literature that used the same method. If it relates to the discussion of IMF% results, then these are discussed extensively in the 1st paragraph of section 3.2.

Reviewer 2 Report

Material and Methods

Lines 10 and 11 on page 3: Detail the meaning of "excess fat removed"

"Briefly, for the grill cook method, cuts were dis-110 sected from the carcass, trimmed of excess fat and denuded of epimysium"

 Results

 Discussion

 General observations: Page 8, line 294 Watson et al., (2008) x Page 9, line 327 Bouton, et al. [38] Ø  General remarks on citation form. What is the norm and why the differences. Ø  Check in all manuscript Ø  Check text formatting: Line 298 The MSA beef eating quality model predicts that roasted Ø  Check norms and standardize across the text (65oC 55 °C); Ø  Check norms and standardize across the text (1°C and frozen at -20 °C) Ø  Check norms and standardize across the text (1cm2) Ø  Check norms and standardize across the text (15 mm thick x 50mm)

Pag 9, lines 301 to 307

" This outcome corroborates the work 301 of [24], who noted that the eating quality of the two species was distinctly different. These 302 differences between lamb and beef might due to differences in age at slaughter, with beef 303 typically slaughtered older than lamb [3,25], as well as the size of the animals and muscles 304 [26,27], and/or nutrition which causes diverse differences within beef, in contrast with 305 lamb [28]. The underlying mechanisms causing the opposite effects in beef and lamb could 306 be a target for future research "

Comment 1: Discuss how age and slaughter weight can be related to your results Comment 2: Discuss based on the literature which nutritional aspects (mechanisms) exist in the two species (bovine x sheep) that can improve your discussion

 Pag 9, lines 318 to 319

"In contrast with the literature, this experiment showed that the 318 MQ4 scores for grilled lamb cuts scored higher for all three muscles than roasted"

Comment: Discuss why your results are opposed to those cited

 Pag 9, lines 348 to 349

"These differences might also relate to water-holding capacity [47] and lipids dis-348 charged during mastication [35]".

Comment: This statement requires a detailing of what was expressed in the material and method (Material and Methods - Lines 10 and 11 on page 3 Detail the meaning of "excess fat removed")

 Conclusions

The conclusion cannot be a repetition of results and discussions. Therefore, redo the conclusion

 References

Conferir todas as referências

12. Pannier, L.; Gardner, G.; Pearce, K.; McDonagh, M.; Ball, A.; Jacob, R.; Pethick, D. Associations of sire estimated breeding values 451 and objective meat quality measurements with sensory scores in Australian lamb. Meat Sci. 2014, 96, 1076-1087.

 27. Pannier, L.; Gardner, G.; O'Reilly, R.; Pethick, D. Factors affecting lamb eating quality and the potential for their integration into 482 an MSA sheepmeat grading model. Meat Sci. 2018, 144, 43-52.

 5. Thompson, J.; Gee, A.; Hopkins, D.; Pethick, D.; Baud, S.; O’Halloran, W. Development of a sensory protocol for testing 438 palatability of sheep meats. Aust. J. Exp. Agric 2005, 45, 469-476.

 8. Thompson, J.; Polkinghorne, R.; Gee, A.; Motiang, D.; Strydom, P.; Mashau, M.; Ng'ambi, J.; Kock, R.d.; Burrow, H. Beef 444 palatability in the Republic of South Africa: implications for niche-marketing strategies. ACIAR Technical Reports Series 2010.

 Author Response

Reviewer #2

1.

Lines 10 and 11 on page 3: Detail the meaning of "excess fat removed"

"Briefly, for the grill cook method, cuts were dis-110 sected from the carcass, trimmed of excess fat and denuded of epimysium"

‘excess fat’ has been changed to ‘subcutaneous and intermuscular fat’

2.

General observations: Page 8, line 294 Watson et al., (2008) x Page 9, line 327 Bouton, et al. [38] Ø  General remarks on citation form. What is the norm and why the differences. Ø  Check in all manuscript Ø  Check text formatting: Line 298 The MSA beef eating quality model predicts that roasted Ø  Check norms and standardize across the text (65oC 55 °C); Ø  Check norms and standardize across the text (1°C and frozen at -20 °C) Ø  Check norms and standardize across the text (1cm2) Ø  Check norms and standardize across the text (15 mm thick x 50mm)

Accepted. Reference formatting was revised throughout the manuscript according to the journal formatting and changes have been made accordingly.

°C used throughout with the same spaces.

3.

Pag 9, lines 301 to 307

" This outcome corroborates the work 301 of [24], who noted that the eating quality of the two species was distinctly different. These 302 differences between lamb and beef might due to differences in age at slaughter, with beef 303 typically slaughtered older than lamb [3,25], as well as the size of the animals and muscles 304 [26,27], and/or nutrition which causes diverse differences within beef, in contrast with 305 lamb [28]. The underlying mechanisms causing the opposite effects in beef and lamb could 306 be a target for future research "

Comment 1: Discuss how age and slaughter weight can be related to your results

More information added

4.

Comment 2: Discuss based on the literature which nutritional aspects (mechanisms) exist in the two species (bovine x sheep) that can improve your discussion

Even though differences between these cut by cook responses exist between beef and lamb, slaughter age or weight had no impact on the grill and roast results within the lamb experiment because paired samples came from the left and right legs of each lamb. Hence the differences between the grill and roast samples from the lambs can not be explained by these factors.  

5.

Pag 9, lines 318 to 319

"In contrast with the literature, this experiment showed that the 318 MQ4 scores for grilled lamb cuts scored higher for all three muscles than roasted"

Comment: Discuss why your results are opposed to those cited

Some more detail has been added to this section for greater clarity of this sentence which relates to the literature on beef eating and not ovine. This is the first publication comparing the 2.   

6

Pag 9, lines 348 to 349

"These differences might also relate to water-holding capacity [47] and lipids dis-348 charged during mastication [35]".

Comment: This statement requires a detailing of what was expressed in the material and method (Material and Methods - Lines 10 and 11 on page 3 Detail the meaning of "excess fat removed")

More clarity has been added at the start of this sentence to allow greater readability.

7

Conclusions

The conclusion cannot be a repetition of results and discussions. Therefore, redo the conclusion

Accepted. Conclusion significantly changed

8.

References

Conferir todas as referências

12. Pannier, L.; Gardner, G.; Pearce, K.; McDonagh, M.; Ball, A.; Jacob, R.; Pethick, D. Associations of sire estimated breeding values 451 and objective meat quality measurements with sensory scores in Australian lamb. Meat Sci. 2014, 96, 1076-1087.

27. Pannier, L.; Gardner, G.; O'Reilly, R.; Pethick, D. Factors affecting lamb eating quality and the potential for their integration into 482 an MSA sheepmeat grading model. Meat Sci. 2018, 144, 43-52.

5. Thompson, J.; Gee, A.; Hopkins, D.; Pethick, D.; Baud, S.; O’Halloran, W. Development of a sensory protocol for testing 438 palatability of sheep meats. Aust. J. Exp. Agric 2005, 45, 469-476.

8. Thompson, J.; Polkinghorne, R.; Gee, A.; Motiang, D.; Strydom, P.; Mashau, M.; Ng'ambi, J.; Kock, R.d.; Burrow, H. Beef 444 palatability in the Republic of South Africa: implications for niche-marketing strategies. ACIAR Technical Reports Series 2010.

Accepted. All references checked and formatting corrected.

Reviewer 3 Report

The paper titled "Quantifying the Effect of Grilling and Roasting on the Eating Quality of Lamb Leg Muscles" presents a comprehensive study that investigates the impact of different cooking methods on the eating quality of lamb leg muscles. The research is well-executed, and the discussions are thoughtfully constructed. The paper has several strengths, however, Tables 1 and 2 should be placed in the results section. Other areas that require improvement, such as the clarity of Figure 1 and Figure 2, and the need to balance the results and discussion sections.

The inclusion of Tables 1 and 2 in the results section greatly enhances the clarity of the presented data.

While Figure 1 and Figure 2 are important visual aids, their clarity needs improvement. The readability of these figures is crucial for conveying the study's findings. The authors should enhance the clarity of these figures by using larger font sizes, clearer labels, and appropriately contrasting colors.

There seems to be an imbalance between the results and discussion sections. While the discussions are well-constructed and informative, the results section might need to be expanded. To maintain a coherent flow, the authors should ensure that the results section provides adequate detail on the methodology used, the raw data obtained, and the statistical analyses performed, before delving into the discussions.

Other Recommendations for Revision:

Line 73, please correct the degree sign.

Place tables 1 and 2 in the results section.

In table 2, min and max values should be separated; to avoid confusion. Why the grill is only 64 samples?

The authors should review Figure 1 and Figure 2 to ensure they are easily readable by a wide audience.

Correct line 143.

Under each table, explain the abbreviations used in the tables.

The statistical model is not very clear. Please make it more easily to be understood and appliable by other readers.

Author Response

Reviewer #3

The paper titled "Quantifying the Effect of Grilling and Roasting on the Eating Quality of Lamb Leg Muscles" presents a comprehensive study that investigates the impact of different cooking methods on the eating quality of lamb leg muscles. The research is well-executed, and the discussions are thoughtfully constructed. The paper has several strengths, however, Tables 1 and 2 should be placed in the results section. Other areas that require improvement, such as the clarity of Figure 1 and Figure 2, and the need to balance the results and discussion sections.

The inclusion of Tables 1 and 2 in the results section greatly enhances the clarity of the presented data.

Accepted. Tables 1-3 were moved to the results section as suggested. The results and discussion sections have also been balanced to have the same flow of ideas.  

2.

While Figure 1 and Figure 2 are important visual aids, their clarity needs improvement. The readability of these figures is crucial for conveying the study's findings. The authors should enhance the clarity of these figures by using larger font sizes, clearer labels, and appropriately contrasting colors.

Accepted. Figures are changed  

3

There seems to be an imbalance between the results and discussion sections. While the discussions are well-constructed and informative, the results section might need to be expanded. To maintain a coherent flow, the authors should ensure that the results section provides adequate detail on the methodology used, the raw data obtained, and the statistical analyses performed, before delving into the discussions.

The authors believe all results were discussed in adequate detail and lengthening the results would detract from the manuscript.

2.

Line 73, please correct the degree sign.

Accepted. The degree sign was corrected as suggested.

3.

Place tables 1 and 2 in the results section.

Accepted. Tables 1-3 were moved to the results section as suggested.

4.

In table 2, min and max values should be separated; to avoid confusion. Why the grill is only 64 samples?

Accepted. The minimum and maximum values were separated into different columns for clarity.

One grill sample was damaged during portioning

5.

The authors should review Figure 1 and Figure 2 to ensure they are easily readable by a wide audience.

Accepted. Figure improved

6.

Correct line 143.

Accepted. Heading 2.2.4 moved to a new line.

7.

Under each table, explain the abbreviations used in the tables.

 Abbreviations in Table 1 were already defined in the table heading. There are no abbreviations in Tables 2 and 3.

8.

The statistical model is not very clear. Please make it more easily to be understood and appliable by other readers.

The authors believe that this description adequately represents the statistical analysis description needed.

Reviewer 4 Report

The maniscript focuses on the lamb meat and the possibility to assess the effect of the cooking method on the eating quality of specific muscles that can be purchased by consumers.  Although the study concerns the meats in Australia, the results are valuable and might be applied worldwide. 

The Introduction of the manuscript is concise but contains the necessary information ( supported by refernces) to introduce the importance and the necessity of the research to the readers. The authors refer to a previously done research and development of prediction models in beef that might be successfully applied for lambs as well. The aim of the study is very clearly defined. 

Material and methods: As a whole the materail and method section is described in details, however in the part of the experimental design, the authors include Table 1 (concerning the cross of the lambs used in the experiment), however the one presented contains completely different data and should be included in the presentation of the results. 

When describing the sample preparation, the authors should use italic for the Latin names (lines 102-104). 

Results: Table 1 that contains data about the fat, sherforce, etc., should be corrected and the units in which each of the tait is measured should be added. In addition, if the authots would like to present the cross of lambs and the respective parent breeds used in a table in Material and methods, they should change the number of the tables accordingly.

The discussion is exhaustive and contains appropriate references for comparison and/ or support of the results of the authors. 

The conclusion are supported by the results. 

The authors should however stick to the instructions about references in the text, because some are not in the reguired format ( lines 316-317; 320-321; 331; 336; 356; 372; 373; 380; 388). 

In the Abstract the auhors should report the real p values, and not p<0.05; P> 0.05.... 

The English language requires minor editing.

Line 83: might be corrected as : Samples (40g) for intramuscular fat (IMF) were collected from the caudal end of the  m. longissimus thoracis (13th thoracic vertebrae) and the cranial end of the LL (1st lumber 84 vertebrae). 

Line 85: The samples were put in freeze dryer or freez dried. 

Author Response

Reviewer #4

1.

Material and methods: As a whole the materail and method section is described in details, however in the part of the experimental design, the authors include Table 1 (concerning the cross of the lambs used in the experiment), however the one presented contains completely different data and should be included in the presentation of the results.

Accepted. Table 1 was moved to the results.

2.

When describing the sample preparation, the authors should use italic for the Latin names (lines 102-104).

Accepted. Latin names were italicized throughout the manuscript.

3.

Results: Table 1 that contains data about the fat, sherforce, etc., should be corrected and the units in which each of the tait is measured should be added. In addition, if the authots would like to present the cross of lambs and the respective parent breeds used in a table in Material and methods, they should change the number of the tables accordingly.

Units have been added to table 1. There isn’t sufficient data to add in sire breed into the model. They were included in the methods because they represent a normal group of lambs.  

4.

The discussion is exhaustive and contains appropriate references for comparison and/ or support of the results of the authors.

Thankyou

5.

The conclusion are supported by the results.

The authors should however stick to the instructions about references in the text, because some are not in the reguired format ( lines 316-317; 320-321; 331; 336; 356; 372; 373; 380; 388).

Accepted. References were revised according to the journal format.

6.

In the Abstract the auhors should report the real p values, and not p<0.05; P> 0.05....

Accepted. P value changed

7.

Line 83: might be corrected as : Samples (40g) for intramuscular fat (IMF) were collected from the caudal end of the  m. longissimus thoracis (13th thoracic vertebrae) and the cranial end of the LL (1st lumber 84 vertebrae).

Accepted. The Sentence was revised as suggested.

8.

Line 85: The samples were put in freeze dryer or freez dried.

This sentence has been corrected.  

Round 2

Reviewer 1 Report

Line99: While this manuscript focuses on consumer appreciation of cooked lamb for tenderness, flavor (taste), juiciness, and overall preference, meat color is an equally important indicator of consumer evaluation of cooked lamb that needs to be added.

Line221: Volatile aromas determined by GC-MS are also required to assess the effect of different cooking methods on the eating quality of lamb.

需要对英语进行少量编辑。

Author Response

Line99: While this manuscript focuses on consumer appreciation of cooked lamb for tenderness, flavor (taste), juiciness, and overall preference, meat color is an equally important indicator of consumer evaluation of cooked lamb that needs to be added.

Whilst the authors acknowledge that color is also an important aspect of consumer product preference at the time of purchasing (Hastie et al., 2020; Kropf, 1980; Tomasevic et al., 2021), color was not measured in the current study and can therefore not be included. This study focuses on cooked meat products and therefore the consumers are not aware of the actual meat color before cooking. Additionally, previous work has looked at the impact of color on the actual eating quality, and color is a poor indicator of the eating quality of a cooked product (Kelly and Thompson, 2014). Furthermore, color measured at the ribeye at grading under MSA is not strongly linked to the color at retail, and color is now an optional trait under beef MSA due to the lack of evidence of its prediction impact on eating quality when other traits like pH are included in the MSA model (Thompson, 2016). However, the latter does not take away the fact that color remains important at the time of purchase.

Regarding color at the time of cooking (i.e. browning), this is highly dependent on the pH of the muscles and the cooking temperature used (Brewer MS and Novakofski J., 1999). Individual muscle pH was not measured and varying temperatures within each cooking method were not tested as this was outside the scope of this study.

Line221: Volatile aromas determined by GC-MS are also required to assess the effect of different cooking methods on the eating quality of lamb.

Whilst the authors agree that volatile aromas are developed during cooking, and may vary between cooking styles, they may play an impact on consumer preference, particularly flavor. However, determining these volatile aromas was outside the scope of this study and these were not measured. The aim was to identify the impact of cut-by-cooking method on untrained consumer sensory scores for the inclusion into a cut-by-cooking method eating quality prediction model through the use of carcass predictors (Pannier et al., 2018). As outlined in the manuscript many other factors can impact eating quality; levels of connective tissue, collagen level, and individual muscle lipid content which is strongly associated with the level of volatiles (Frank et al., 2016). However, these were not measured as they are all imbedded in the overall cut by cooking description defining the eating quality as a whole.

Comments on the Quality of English Language

需要对英语进行少量编辑。

The manuscript has been checked again and reviewed by native English-speaking authors, as such the authors are happy to address specific English queries that the reviewer highlights.

References

Kropf, D. (1980). Effects of retail display conditions on meat color. Proceedings Annual Reciprocal Meat Conference, 15-32.

Hastie, M., Ashman, H., Torrico, D., Ha, M., & Warner, R. (2020). A Mixed Method Approach for the Investigation of Consumer Responses to Sheepmeat and Beef. Foods, 9(2), 126.

Tomasevic, I., Djekic, I., Font-i-Furnols, M., Terjung, N., & Lorenzo, J. M. (2021). Recent advances in meat color research. Current Opinion in Food Science, 41, 81-87.

Kelly MJ and Thompson JM (2014) Utilising genetic markers to improve the understanding of the relationship between Bos indicus content and consumer eating quality. Final Report Project V.EQT.1420. Meat and Livestock Australia, Sydney

Thompson, J. M. (2016). Current carcase traits in the Australian beef language. In Technical papers for the Australian beef language “white paper”. Sydney, Australia: Meat and Livestock Australia.

Brewer MS and Novakofski J. (1999). Cooking rate, pH and final endpoint temperature effects on color and cook loss of a lean ground beef model system, Meat Science,Volume 52, Issue 4

Pannier, L.; Gardner, G.; O'Reilly, R.; Pethick, D. Factors affecting lamb eating quality and the potential for their integration into an MSA sheepmeat grading model. Meat Sci. 2018, 144, 43-52.

Frank, D., Ball, A., Hughes, J., Krishnamurthy, R., Piyasiri, U., Stark, J., ... Warner, R. (2016). Sensory and flavor chemistry characteristics of Australian beef: Influence of intramuscular fat, feed, and breed. Journal of Agricultural and Food Chemistry, 64(21), 4299–4311.